# Unsupervised Text Style Transfer using Language Models as Discriminators

**Zichao Yang**[1], **Zhiting Hu**[1], **Chris Dyer**[2], **Eric P. Xing**[1], **Taylor Berg-Kirkpatrick**[1]
[1]Carnegie Mellon University, [2]DeepMind
{zichaoy, zhitingh, epxing, tberg}@cs.cmu.edu
cdyer@google.com

## Abstract

Binary classifiers are often employed as discriminators in GAN-based unsupervised style transfer systems to ensure that transferred sentences are similar to sentences in the target domain. One difficulty with this approach is that the error signal provided by the discriminator can be unstable and is sometimes insufficient to train the generator to produce fluent language. In this paper, we propose a new technique that uses a target domain language model as the discriminator, providing richer and more stable token-level feedback during the learning process. We train the generator to minimize the negative log likelihood (NLL) of generated sentences, evaluated by the language model. By using a continuous approximation of discrete sampling under the generator, our model can be trained using back-propagation in an end-to-end fashion. Moreover, our empirical results show that when using a language model as a structured discriminator, it is possible to forgo adversarial steps during training, making the process more stable. We compare our model with previous work that uses convolutional networks (CNNs) as discriminators, as well as a broad set of other approaches. Results show that the proposed method achieves improved performance on three tasks: word substitution decipherment, sentiment modification, and related language translation.

## 1   Introduction

Recently there has been growing interest in designing natural language generation (NLG) systems that allow for control over various attributes of generated text – for example, sentiment and other stylistic properties. Such controllable NLG models have wide applications in dialogues systems (Wen et al., 2016) and other natural language interfaces. Recent successes for neural text generation models in machine translation (Bahdanau et al., 2014), image captioning (Vinyals et al., 2015) and dialogue (Vinyals and Le, 2015; Wen et al., 2016) have relied on massive parallel data. However, for many other domains, only non-parallel data – which includes collections of sentences from each domain without explicit correspondence – is available. Many text style transfer problems fall into this category. The goal for these tasks is to transfer a sentence with one attribute to a sentence with an another attribute, but with the same style-independent content, trained using only non-parallel data.

Unsupervised text style transfer requires learning disentangled representations of attributes (e.g., negative/positive sentiment, plaintext/ciphertext orthography) and underlying content. This is challenging because the two interact in subtle ways in natural language and it can even be hard to disentangle them with parallel data. The recent development of deep generative models like variational auto-encoders (VAEs) (Kingma and Welling, 2013) and generative adversarial networks(GANs) (Goodfellow et al., 2014) have made learning disentangled representations from non-parallel data possible. However, despite their rapid progress in computer vision—for example, generating photo-realistic images (Radford et al., 2015), learning interpretable representations (Chen et al., 2016b), and translating im-

ages (Zhu et al., 2017)—their progress on text has been more limited. For VAEs, the problem of training collapse can severely limit effectiveness (Bowman et al., 2015; Yang et al., 2017b), and when applying adversarial training to natural language, the non-differentiability of discrete word tokens makes generator optimization difficult. Hence, most attempts use REINFORCE (Sutton et al., 2000) to finetune trained models (Yu et al., 2017; Li et al., 2017) or uses professor forcing (Lamb et al., 2016) to match hidden states of decoders.

Previous work on unsupervised text style transfer (Hu et al., 2017a; Shen et al., 2017) adopts an encoder-decoder architecture with style discriminators to learn disentangled representations. The encoder takes a sentence as an input and outputs a style-independent content representation. The style-dependent decoder takes the content representation and a style representation and generates the transferred sentence. Hu et al. (2017a) use a style classifier to directly enforce the desired style in the generated text. Shen et al. (2017) leverage an adversarial training scheme where a binary CNN-based discriminator is used to evaluate whether a transferred sentence is real or fake, ensuring that transferred sentences match real sentences in terms of target style. However, in practice, the error signal from a binary classifier is sometimes insufficient to train the generator to produce fluent language, and optimization can be unstable as a result of the adversarial training step.

We propose to use an implicitly trained language model as a new type of discriminator, replacing the more conventional binary classifier. The language model calculates a sentence's likelihood, which decomposes into a product of token-level conditional probabilities. In our approach, rather than training a binary classifier to distinguish real and fake sentences, we train the language model to assign a high probability to real sentences and train the generator to produce sentences with high probability under the language model. Because the language model scores sentences directly using a product of locally normalized probabilities, it may offer more stable and more useful training signal to the generator. Further, by using a continuous approximation of discrete sampling under the generator, our model can be trained using back-propagation in an end-to-end fashion.

We find empirically that when using the language model as a structured discriminator, it is possible to eliminate adversarial training steps that use negative samples—a critical part of traditional adversarial training. Language models are *implicitly* trained to assign a low probability to negative samples because of its normalization constant. By eliminating the adversarial training step, we found the training becomes more stable in practice.

To demonstrate the effectiveness of our new approach, we conduct experiments on three tasks: word substitution decipherment, sentiment modification, and related language translation. We show that our approach, which uses only a language model as the discriminator, outperforms a broad set of state-of-the-art approaches on the three tasks.

## 2    Unsupervised Text Style Transfer

We start by reviewing the current approaches for unsupervised text style transfer (Hu et al., 2017a; Shen et al., 2017), and then go on to describe our approach in Section 3. Assume we have two text datasets $\mathbf{X} = \{\mathbf{x}^{(1)}, \mathbf{x}^{(2)}, \ldots, \mathbf{x}^{(m)}\}$ and $\mathbf{Y} = \{\mathbf{y}^{(1)}, \mathbf{y}^{(2)}, \ldots, \mathbf{y}^{(n)}\}$ with two different styles $\mathbf{v}_x$ and $\mathbf{v}_y$, respectively. For example, $\mathbf{v}_x$ can be the positive sentiment style and $\mathbf{v}_y$ can be the negative sentiment style. The datasets are non-parallel such that the data does not contain pairs of $(\mathbf{x}^{(i)}, \mathbf{y}^{(j)})$ that describe the same content. The goal of style transfer is to transfer data $\mathbf{x}$ with style $\mathbf{v_x}$ to style $\mathbf{v_y}$ and vice versa, i.e., to estimate the conditional distribution $p(\mathbf{y}|\mathbf{x})$ and $p(\mathbf{x}|\mathbf{y})$. Since text data is discrete, it is hard to learn the transfer function directly via back-propagation as in computer vision (Zhu et al., 2017). Instead, we assume the data is generated conditioned on two disentangled parts, the style $\mathbf{v}$ and the content $\mathbf{z}$[1] (Hu et al., 2017a).

Consider the following generative process for each style: 1) the style representation $\mathbf{v}$ is sampled from a prior $p(\mathbf{v})$; 2) the content vector $\mathbf{z}$ is sampled from $p(\mathbf{z})$; 3) the sentence $\mathbf{x}$ is generated from the conditional distribution $p(\mathbf{x}|\mathbf{z}, \mathbf{v})$. This model suggests the following parametric form for style transfer where $q$ represents a posterior:

$$p(\mathbf{y}|\mathbf{x}) = \int_{\mathbf{z_x}} p(\mathbf{y}|\mathbf{z_x}, \mathbf{v_y}) q(\mathbf{z_x}|\mathbf{x}, \mathbf{v_x}) d\mathbf{z_x}.$$

The above equation suggests the use of an encoder-decoder framework for style transfer problems. We can first encode the sentence $\mathbf{x}$ to get its content vector $\mathbf{z_x}$, then we switch the style label from $\mathbf{v_x}$ to $\mathbf{v_y}$. Combining the content vector $\mathbf{z_x}$ and the style label $\mathbf{v_y}$, we can generate a new sentence $\tilde{\mathbf{x}}$ (the transferred sentences are denotes as $\tilde{\mathbf{x}}$ and $\tilde{\mathbf{y}}$).

One unsupervised approach is to use the auto-encoder model. We first use an encoder model $\mathbf{E}$ to encode $\mathbf{x}$ and $\mathbf{y}$ to get the content vectors $\mathbf{z_x} = \mathbf{E}(\mathbf{x}, \mathbf{v_x})$ and $\mathbf{z_y} = \mathbf{E}(\mathbf{y}, \mathbf{v_y})$. Then we use a decoder $\mathbf{G}$ to generate sentences conditioned on $\mathbf{z}$ and $\mathbf{v}$. The $\mathbf{E}$ and $\mathbf{G}$ together form an auto-encoder and the reconstruction loss is:

$$\mathcal{L}_{\text{rec}}(\theta_{\mathbf{E}}, \theta_{\mathbf{G}}) = \mathbb{E}_{\mathbf{x} \sim \mathbf{X}}[-\log p_{\mathbf{G}}(\mathbf{x}|\mathbf{z_x}, \mathbf{v_x})] + \mathbb{E}_{\mathbf{y} \sim \mathbf{Y}}[-\log p_{\mathbf{G}}(\mathbf{y}|\mathbf{z_y}, \mathbf{v_y})],$$

where $\mathbf{v_x}$ and $\mathbf{v_y}$ can be two learnable vectors to represent the label embedding. In order to make sure that the $\mathbf{z_x}$ and $\mathbf{z_y}$ capture the content and we can deliver accurate transfer between the style by switching the labels, we need to guarantee that $\mathbf{z_x}$ and $\mathbf{z_y}$ follow the same distribution. We can assume $p(\mathbf{z})$ follows a prior distribution and add a KL-divergence regularization on $\mathbf{z_x}$, $\mathbf{z_y}$. The model then becomes a VAE. However, previous works (Bowman et al., 2015; Yang et al., 2017b) found that there is a training collapse problem with the VAE for text modeling and the posterior distribution of $\mathbf{z}$ fails to capture the content of a sentence.

To better capture the desired styles in the generated sentences, Hu et al. (2017a) additionally impose a style classifier on the generated samples, and the decoder $\mathbf{G}$ is trained to generate sentences that maximize the accuracy of the style classifier. Such additional supervision with a *discriminative* model is also adopted in (Shen et al., 2017), though in that work a binary real/fake classifier is instead used within a conventional adversarial scheme.

**Adversarial Training**  Shen et al. (2017) use adversarial training to align the $\mathbf{z}$ distributions. Not only do we want to align the distribution of $\mathbf{z_x}$ and $\mathbf{z_y}$, but also we hope that the transferred sentence $\tilde{\mathbf{x}}$ from $\mathbf{x}$ to resemble $\mathbf{y}$ and vice versa. Several adversarial discriminators are introduced to align these distributions. Each of the discriminators is a binary classifier distinguishing between real and fake. Specifically, the discriminator $D_{\mathbf{z}}$ aims to distinguish between $\mathbf{z_x}$ and $\mathbf{z_y}$:

$$\mathcal{L}_{\text{adv}}^{\mathbf{z}}(\theta_{\mathbf{E}}, \theta_{\mathbf{D_z}}) = \mathbb{E}_{\mathbf{x} \sim \mathbf{X}}[-\log D_{\mathbf{z}}(\mathbf{z_x})] + \mathbb{E}_{\mathbf{y} \sim \mathbf{Y}}[-\log(1 - D_{\mathbf{z}}(\mathbf{z_y}))].$$

Similarly, $D_{\mathbf{x}}$ distinguish between $\mathbf{x}$ and $\tilde{\mathbf{y}}$, yielding an objective $\mathcal{L}_{\text{adv}}^{\mathbf{x}}$ as above; and $D_{\mathbf{y}}$ distinguish between $\mathbf{y}$ and $\tilde{\mathbf{x}}$, yielding $\mathcal{L}_{\text{adv}}^{\mathbf{y}}$. Since the samples of $\tilde{\mathbf{x}}$ and $\tilde{\mathbf{y}}$ are discrete and it is hard to train the generator in an end-to-end way, professor forcing (Lamb et al., 2016) is used to match the distributions of the hidden states of decoders. The overall training objective is a min-max game played among the encoder $\mathbf{E}$/decoder $\mathbf{G}$ and the discriminators $D_{\mathbf{z}}, D_{\mathbf{x}}, D_{\mathbf{y}}$ (Goodfellow et al., 2014):

$$\min_{E, G} \max_{D_{\mathbf{z}}, D_{\mathbf{x}}, D_{\mathbf{y}}} \mathcal{L}_{\text{rec}} - \lambda(\mathcal{L}_{\text{adv}}^{\mathbf{z}} + \mathcal{L}_{\text{adv}}^{\mathbf{x}} + \mathcal{L}_{\text{adv}}^{\mathbf{y}})$$

The model is trained in an alternating manner. In the first step, the loss of the discriminators are minimize to distinguish between the $\mathbf{z_x}, \mathbf{x}, \mathbf{y}$ and $\mathbf{z_y}, \tilde{\mathbf{x}}, \tilde{\mathbf{y}}$, respectively; and in the second step the encoder and decoder are trained to minimize the reconstruction loss while maximizing loss of the discriminators.

## 3  Language Models as Discriminators

In most past work, a classifier is used as the discriminator to distinguish whether a sentence is real or fake. We propose instead to use locally-normalized language models as discriminators. We argue that using an explicit language model with token-level locally normalized probabilities offers a more direct training signal to the generator. If a transfered sentence does not match the target style, it will have high perplexity when evaluated by a language model that was trained on target domain data. Not only does it provide an overall evaluation score for the whole sentence, but a language model can also assign a probability to each token, thus providing more information on which word is to blame if the overall perplexity is very high.

The overall model architecture is shown in Figure 1. Suppose $\tilde{\mathbf{x}}$ is the output sentence from applying style transfer to input sentence $\mathbf{x}$, i.e., $\tilde{\mathbf{x}}$ is sampled from $p_G(\tilde{\mathbf{x}}|\mathbf{z_x}, \mathbf{v_y})$ (and similary for $\tilde{\mathbf{y}}$ and $\mathbf{y}$). Let $p_{\text{LM}}(\mathbf{x})$ be the probability of a sentence $\mathbf{x}$ evaluate against a language model, then the discriminator

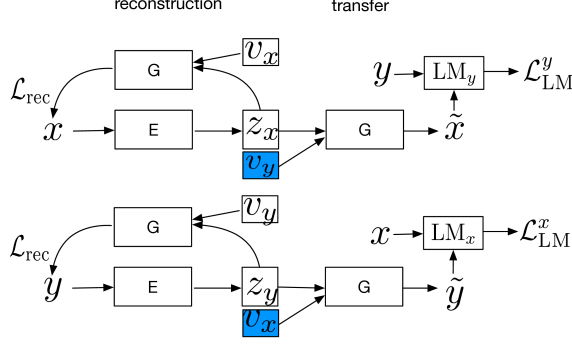

Figure 1: The overall model architecture consists of two parts: reconstruction and transfer. For transfer, we switch the style label and sample an output sentence from the generator that is evaluated by a language model.

loss becomes:

$$\mathcal{L}_{\text{LM}}^{\mathbf{x}}(\theta_{\mathbf{E}}, \theta_{\mathbf{G}}, \theta_{\text{LM}_{\mathbf{x}}}) = \mathbb{E}_{\mathbf{x} \sim \mathbf{X}}[- \log p_{\text{LM}_{\mathbf{x}}}(\mathbf{x}))] + \gamma \mathbb{E}_{\mathbf{y} \sim \mathbf{Y}, \tilde{\mathbf{y}} \sim p_G(\tilde{\mathbf{y}}|\mathbf{z_y}, \mathbf{v_x})}[\log p_{\text{LM}_{\mathbf{x}}}(\tilde{\mathbf{y}})], \quad (1)$$

$$\mathcal{L}_{\text{LM}}^{\mathbf{y}}(\theta_{\mathbf{E}}, \theta_{\mathbf{G}}, \theta_{\text{LM}_{\mathbf{y}}}) = \mathbb{E}_{\mathbf{y} \sim \mathbf{Y}}[- \log p_{\text{LM}_{\mathbf{y}}}(\mathbf{y}))] + \gamma \mathbb{E}_{\mathbf{x} \sim \mathbf{X}, \tilde{\mathbf{x}} \sim p_G(\tilde{\mathbf{x}}|\mathbf{z_x}, \mathbf{v_y})}[\log p_{\text{LM}_{\mathbf{y}}}(\tilde{\mathbf{x}})]. \quad (2)$$

Our overall objective becomes:

$$\min_{E,G} \max_{\text{LM}_{\mathbf{x}}, \text{LM}_{\mathbf{y}}} \mathcal{L}_{\text{rec}} - \lambda(\mathcal{L}_{\text{LM}}^{\mathbf{x}} + \mathcal{L}_{\text{LM}}^{\mathbf{y}}) \quad (3)$$

**Negative samples**: Note that Equation 1 and 2 differs from traditional ways of training language models in that we have a term including the negative samples. We train the LM in an adversarial way by minimizing the loss of LM of real sentences and maximizing the loss of transferred sentences. However, since the LM is a structured discriminator, we would hope that a language model trained on the real sentences will automatically assign high perplexity to sentences not in the target domain, hence negative samples from the generator may not be necessary. To investigate the necessity of negative samples, we add a weight $\gamma$ to the loss of negative samples. The weight $\gamma$ adjusts the negative sample loss in training the language models. If $\gamma = 0$, we simply train the language model on real sentences and fix its parameters, avoiding potentially unstable adversarial training steps. We investigate the necessity of using negative samples in the experiment section.

Training consists of two steps alternatively. In the first step, we train the language models according to Equation 1 and 2. In the second step, we minimize the reconstruction loss as well as the perplexity of generated samples evaluated by the language model. Since $\tilde{\mathbf{x}}$ is discrete, one can use the REINFORCE (Sutton et al., 2000) algorithm to train the generator:

$$\nabla_{\theta_G} \mathcal{L}_{\text{LM}}^{\mathbf{y}} = \mathbb{E}_{\mathbf{x} \sim \mathbf{X}, \tilde{\mathbf{x}} \sim p_G(\tilde{\mathbf{x}}|\mathbf{z_x}, \mathbf{v_y})}[\log p_{\text{LM}}(\tilde{\mathbf{x}}) \nabla_{\theta_G} \log p_G(\tilde{\mathbf{x}}|\mathbf{z_x}, \mathbf{v_y})]. \quad (4)$$

However, using a single sample to approximate the expected gradient leads to high variance in gradient estimates and thus unstable learning.

**Continuous approximation**: Instead, we propose to use a continuous approximation to the sampling process in training the generator, as demonstrated in Figure 2. Instead of feeding a single sampled word as input to the next timestep of the generator, we use a Gumbel-softmax (Jang et al., 2016) distribution as a continuous approximation to sample instead. Let $u$ be a categorical distribution with probabilities $\pi_1, \pi_2, \ldots, \pi_c$. Samples from $u$ can be approximated using:

$$p_i = \frac{\exp((\log \pi_i) + g_i)/\tau)}{\sum_{j=1}^{c} \exp((\log \pi_j + g_j)/\tau)},$$

where the $g_i$'s are independent samples from $\text{Gumbel}(0, 1)$.

Let the tokens of the transferred sentence be $\tilde{\mathbf{x}} = \{\tilde{x}_t\}_{t=1}^{T}$. Suppose the output of the logit at timestep $t$ is $\mathbf{v}_t^{\mathbf{x}}$, then $\tilde{\mathbf{p}}_t^{\mathbf{x}} = \text{Gumbel-softmax}(\mathbf{v}_t^{\mathbf{x}}, \tau)$, where $\tau$ is the temperature. When $\tau \to 0$, $\tilde{\mathbf{p}}_t^{\mathbf{x}}$ becomes the one hot representation of token $\tilde{x}_t$. Using the continuous approximation, then the output of the decoder becomes a sequence of probability vectors $\tilde{\mathbf{p}}^{\mathbf{x}} = \{\tilde{\mathbf{p}}_t^{\mathbf{x}}\}_{t=1}^{T}$.

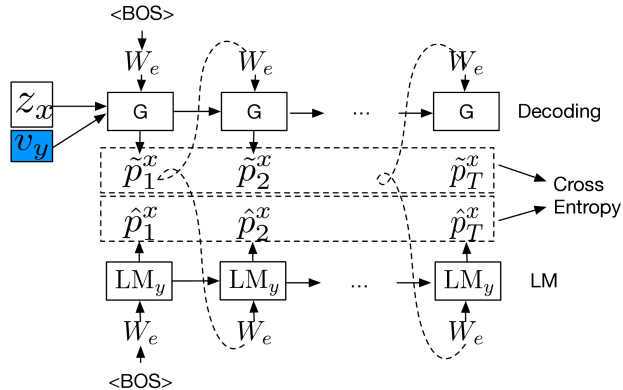

Figure 2: Continuous approximation of language model loss. The input is a sequence of probability distributions $\{\tilde{\mathbf{p}}_t^{\mathbf{x}}\}_{t=1}^T$ sampled from the generator. At each timestep, we compute a weighted embedding as input to the language model and get the sequence of output distributions from the LM as $\{\hat{\mathbf{p}}_t^{\mathbf{x}}\}_{t=1}^T$. The loss is the sum of cross entropies between each pair of $\tilde{\mathbf{p}}_t^{\mathbf{x}}$ and $\hat{\mathbf{p}}_t^{\mathbf{x}}$.

With the continuous approximation of $\tilde{\mathbf{x}}$, we can calculate the loss evaluated using a language model easily, as shown in Figure 2. For every step, we feed $\tilde{\mathbf{p}}_t^{\mathbf{x}}$ to the language model of $\mathbf{y}$ (denoted as $\text{LM}_{\mathbf{y}}$) using the weighted average of the embedding $W_e \tilde{\mathbf{p}}_t^{\mathbf{x}}$, then we get the output from the $\text{LM}_{\mathbf{y}}$ which is a probability distribution over the vocabulary of the next word $\hat{\mathbf{p}}_{t+1}^{\mathbf{x}}$. The loss of the current step is the cross entropy loss between $\tilde{\mathbf{p}}_{t+1}^{\mathbf{x}}$ and $\hat{\mathbf{p}}_{t+1}^{\mathbf{x}}$: $(\tilde{\mathbf{p}}_{t+1}^{\mathbf{x}})^{\mathsf{T}} \log \hat{\mathbf{p}}_{t+1}^{\mathbf{x}}$. Note that when the decoder output distribution $\tilde{\mathbf{p}}_{t+1}^{\mathbf{x}}$ aligns with the language model output distribution $\hat{\mathbf{p}}_{t+1}^{\mathbf{x}}$, the above loss achieves minimum. By summing the loss over all steps and taking the gradient, we can use standard back-propagation to train the generator:

$$\nabla_{\theta_G} \mathcal{L}_{\text{LM}}^{\mathbf{y}} \approx \mathbb{E}_{\mathbf{x} \sim \mathbf{X}, \tilde{\mathbf{p}}^{\mathbf{x}} \sim p_G(\tilde{\mathbf{x}}|\mathbf{z}_{\mathbf{x}}, \mathbf{v}_{\mathbf{y}})}[\nabla_{\theta_G} \sum_{t=1}^T (\tilde{\mathbf{p}}_t^{\mathbf{x}})^{\mathsf{T}} \log \hat{\mathbf{p}}_t^{\mathbf{x}}]. \tag{5}$$

The above Equation is a continuous approximation of Equation 4 with Gumbel softmax distribution. In experiments, we use a single sample of $\tilde{\mathbf{p}}^{\mathbf{x}}$ to approximate the expectation.

Note that the use of the language model discriminator is a somewhat different in each of the two types of training update steps because of the continuous approximation. We use discrete samples from the generators as negative samples in training the language model discriminator step, while we use a continuous approximation in updating the generator step according to Equation 5.

**Overcoming mode collapse**: It is known that in adversarial training, the generator can suffer from mode collapse (Arjovsky and Bottou, 2017; Hu et al., 2017b) where the samples from the generator only cover part of the data distribution. In preliminary experimentation, we found that the language model prefers short sentences. To overcome this length bias, we use two tricks in our experiments: 1) we normalize the loss of Equation 5 by length and 2) we fix the length of $\tilde{\mathbf{x}}$ to be the same of $\mathbf{x}$. We find these two tricks stabilize the training and avoid generating collapsed overly short outputs.

## 4 Experiments

In order to verify the effectiveness of our model, we experiment on three tasks: word substitution decipherment, sentiment modification, and related language translation. We mainly compare with the most comparable approach of (Shen et al., 2017) that uses CNN classifiers as discriminators[2]. Note that Shen et al. (2017) use three discriminators to align both $\mathbf{z}$ and decoder hidden states, while our model only uses a single language model as a discriminator directly on the output sentences $\tilde{\mathbf{x}}, \tilde{\mathbf{y}}$. Moreover, we also compare with a broader set of related work (Hu et al., 2017a; Fu et al., 2017; Li et al., 2018) for the tasks when appropriate. Our proposed model provides substantiate improvements in most of the cases. We implement our model with the Texar (Hu et al., 2018b) toolbox based on Tensorflow (Abadi et al., 2016).

| Model | 20% | 40% | 60% | 80% | 100% |
|---|---|---|---|---|---|
| Copy | 64.3 | 39.1 | 14.4 | 2.5 | 0 |
| Shen et al. (2017)* | 86.6 | 77.1 | 70.1 | 61.2 | **50.8** |
| Our results: | | | | | |
| LM | 89.0 | **80.0** | **74.1** | 62.9 | 49.3 |
| LM + adv | **89.1** | 79.6 | 71.8 | **63.8** | 44.2 |

Table 1: Decipherment results measured in BLEU. Copy is directly measuring **y** against **x**. LM + adv denotes we use negative samples to train the language model.*We run the code open-sourced by the authors to get the results.

| Model | Accu | BLEU | $PPL_X$ | $PPL_Y$ |
|---|---|---|---|---|
| Shen et al. (2017) | 79.5 | 12.4 | 50.4 | 52.7 |
| Hu et al. (2017a) | 87.7 | **65.6** | 115.6 | 239.8 |
| Our results: | | | | |
| LM | 83.3 | 38.6 | **30.3** | **42.1** |
| LM + Classifier | **91.2** | 57.8 | 47.0 | 60.9 |

Table 2: Results for sentiment modification. $X$ = negative, $Y$ = positive. $PPL_x$ denotes the perplexity of sentences transferred from positive sentences evaluated by a language model trained with negative sentences and vice versa.

## 4.1 Word substitution decipherment

As the first task, we consider the word substitution decipherment task previous explored in the NLP literature (Dou and Knight, 2012). We can control the amount of change to the original sentences in word substitution decipherment so as to systematically investigate how well the language model performs in a task that requires various amount of changes. In word substitution cipher, every token in the vocabulary is mapped to a cipher token and the tokens in sentences are replaced with cipher tokens according to the cipher dictionary. The task of decipherment is to recover the original text without any knowledge of the dictionary.

**Data**: Following (Shen et al., 2017), we sample 200K sentences from the Yelp review dataset as plain text **X** and sample other 200K sentences and apply word substitution cipher on these sentences to get **Y**. We use another 100k *parallel* sentences as the development and test set respectively. Sentences of length more than 15 are filtered out. We keep all words that appear more than 5 times in the training set and get a vocabulary size of about 10k. All words appearing less than 5 times are replaced with a "<unk>" token. We random sample words from the vocabulary and replace them with cipher tokens. The amount of ciphered words ranges from 20% to 100%. As we have ground truth plain text, we can directly measure the BLEU [3] score to evaluate the model. Our model configurations are included in Appendix B.

**Results**: The results are shown in Table 1. We first investigate the effect of using negative samples in training the language model, as denotes by LM + adv in Table 1. We can see that using adversarial training sometimes improves the results. However, we found empirically that using negative samples makes the training very unstable and the model diverges easily. This is the main reason why we did not get consistently better results by incorporating adversarial training.

Comparing with (Shen et al., 2017), we can see that the language model without adversarial training is already very effective and performs much better when the amount of change is less than 100%. This is intuitive because when the change is less than 100%, a language model can use context information to predict and correct enciphered tokens. It's surprising that even with 100% token change, our model is only 1.5 BLEU score worse than (Shen et al., 2017), when all tokens are replaced and no context information can be used by the language model. We guess our model can gradually decipher tokens from the beginning of a sentence and then use them as a bootstrap to decipher the whole sentence. We can also combine language models with the CNNs as discriminators. For example, for the 100%

case, we get BLEU score of 52.1 when combing them. Given unstableness of adversarial training and effectiveness of language models, we set $\gamma = 0$ in Equation 1 and 2 in the rest of the experiments.

## 4.2 Sentiment Manipulation

We have demonstrated that the language model can successfully crack word substitution cipher. However, the change of substitution cipher is limited to a one-to-one mapping. As the second task, we would like to investigate whether a language model can distinguish sentences with positive and negative sentiments, thus help to transfer the sentiments of sentences while preserving the content. We compare to the model of (Hu et al., 2017a) as an additional baseline, which uses a pre-trained classifier as guidance.

**Data**: We use the same data set as in (Shen et al., 2017). The data set contains 250K negative sentences (denoted as **X**) and 380K positive sentences (denoted as **Y**), of which 70% are used for training, 10% are used for development and the remaining 20% are used as test set. The pre-processing steps are the same as the previous experiment. We also use similar experiment configurations.

**Evaluation**: Evaluating the quality of transferred sentences is a challenging problem as there are no ground truth sentences. We follow previous papers in using model-based evaluation. We measure whether transferred sentences have the correct sentiment according to a pre-trained sentiment classifier. We follow both (Hu et al., 2017a) and (Shen et al., 2017) in using a CNN-based classifier. However, simply evaluating the sentiment of sentences is not enough since the model can output collapsed output such as a single word "good" for all negative transfer and "bad" for all positive transfer. We not only would like transferred sentences to preserve the content of original sentences, but also to be smooth in terms of language quality. For these two aspects, we propose to measure the BLEU score of transferred sentences against original sentences and measure the perplexity of transferred sentences to evaluate the fluency. A good model should perform well on all three metrics.

**Results**: We report the results in Table. 2. As a baseline, the original corpus has perplexity of 35.8 and 38.8 for the negative and positive sentences respectively. Comparing LM with (Shen et al., 2017), we can see that LM outperforms it in all three aspects: getting higher accuracy, preserving the content better while being more fluent. This demonstrates the effectiveness of using LM as the discriminator. (Hu et al., 2017a) has the highest accuracy and BLEU score among the three models while the perplexity is very high. It is not surprising that the classifier will only modify the features of the sentences that are related to the sentiment and there is no mechanism to ensure that the modified sentence being fluent. Hence the corresponding perplexity is very high. We can manifest the best of both models by combing the loss of LM and the classifier in (Hu et al., 2017a): a classifier is good at modifying the sentiment and an LM can smooth the modification to get a fluent sentence. We find improvement of accuracy and perplexity as denoted by LM + classifier compared to classifier only (Hu et al., 2017a).

**Comparing with other models**: Recently there are other models that are proposed specifically targeting the sentiment modification task such as (Li et al., 2018). Their method is feature based and consists of the following steps: (`Delete`) first, they use the statistics of word frequency to delete the attribute words such as "good, bad" from original sentences, (`Retrieve`) then they retrieve the most similar sentences from the other corpus based on nearest neighbor search, (`Generate`) the attribute words from retrieved sentences are combined with the content words of original sentences to generate transferred sentences. The authors provide 500 human annotated sentences as the ground truth of transferred sentences so we measure the BLEU score against those sentences. The results are shown in Table 3. We can see our model has similar accuracy compared with DeleteAndRetrieve, but has much better BLEU scores and slightly better perplexity.

We list some examples of transferred sentences in Table 5 in the appendix. We can see that (Shen et al., 2017) does not keep the content of the original sentences well and changes the meaning of the original sentences. (Hu et al., 2017a) changes the sentiment but uses improper words, e.g. "maintenance is equally `hilarious`". Our LM can change the change the sentiment of sentences. But sometimes there is an over-smoothing problem, changing the less frequent words to more frequent words, e.g. changing "my goodness it was so gross" to "my `food` it was so good.". In general LM + classifier has the best results, it changes the sentiment, while keeps the content and the sentences are fluent.

| Model | ACCU | BLEU | $PPL_X$ | $PPL_Y$ |
|---|---|---|---|---|
| Shen et al. (2017) | 76.2 | 6.8 | 49.4 | 45.6 |
| Fu et al. (2017): | | | | |
| StyleEmbedding | 9.2 | 16.65 | 97.51 | 142.6 |
| MultiDecoder | 50.9 | 11.24 | 111.1 | 119.1 |
| Li et al. (2018): | | | | |
| Delete | 87.2 | 11.5 | 75.2 | 68.7 |
| Template | 86.7 | 18.0 | 192.5 | 148.4 |
| Retrieval | **95.1** | 1.3 | **31.5** | **37.0** |
| DeleteAndRetrieval | 90.9 | 12.6 | 104.6 | 43.8 |
| Our results: | | | | |
| LM | 85.4 | 13.4 | 32.8 | 40.5 |
| LM + Classifier | 90.0 | **22.3** | 48.4 | 61.6 |

Table 3: Results for sentiment modification based on the 500 human annotated sentences as ground truth from (Li et al., 2018).

### 4.3 Related language translation

In the final experiment, we consider a more challenging task: unsupervised related language translation (Pourdamghani and Knight, 2017). Related language translation is easier than normal pair language translation since there is a close relationship between the two languages. Note here we don't compare with other sophisticated unsupervised neural machine translation systems such as (Lample et al., 2017; Artetxe et al., 2017), whose models are much more complicated and use other techniques such as back-translation, but simply compare the different type of discriminators in the context of a simple model.

**Data**: We choose Bosnian (bs) vs Serbian (sr) and simplified Chinese (zh-CN) vs traditional Chinese (zh-TW) pair as our experiment languages. Due to the lack of parallel data for these data, we build the data ourselves. For bs and sr pair, we use the monolingual data from Leipzig Corpora Collections[4]. We use the news data and sample about 200k sentences of length less than 20 for each language, of which 80% are used for training, 10% are used for validation and remaining 10% are used for test. For validation and test, we obtain the parallel corpus by using the Google Translation API. The vocabulary size is 25k for the sr vs bs language pair. For zh-CN and zh-TW pair, we use the monolingual data from the Chinese Gigaword corpus. We use the news headlines as our training data. 300k sentences are sampled for each language. The data is partitioned and parallel data is obtained in a similar way to that of sr vs bs pair. We directly use a character-based model and the total vocabulary size is about 5k. For evaluation, we directly measure the BLEU score using the references for both language pairs.

Note that the relationship between zh-CN and zh-TW is simple and mostly like a decipherment problem in which some simplified Chinese characters have the corresponding traditional character mapping. The relation between bs vs sr is more complicated.

**Results**: The results are shown in Table. 4. For sr–bos and bos–sr, since the vocabulary of two languages does not overlap at all, it is a very challenging task. We report the BLEU1 metric since the BLEU4 is close to 0. We can see that our language model discriminator still outperforms (Shen et al., 2017) slightly. The case for zh–tw and tw–zh is much easier. Simple copying already has a reasonable score of 32.3. Using our model, we can improve it to 81.6 for cn–tw and 85.5 for tw–cn, outperforming (Shen et al., 2017) by a large margin.

## 5 Related Work

**Non-parallel transfer in natural language**: (Hu et al., 2017a; Shen et al., 2017; Prabhumoye et al., 2018; Gomez et al., 2018) are most relevant to our work. Hu et al. (2017a) aim to generate sentences with controllable attributes by learning disentangled representations. Shen et al. (2017) introduce adversarial training to unsupervised text style transfer. They apply discriminators both on the encoder

| Model | sr–bs | bs–sr | cn–tw | tw–cn |
|---|---|---|---|---|
| Copy | 0 | 0 | 32.3 | 32.3 |
| Shen et al. (2017) | 29.1 | 30.3 | 60.1 | 60.7 |
| Our results: | | | | |
| LM | **31.0** | **31.7** | **81.6** | **85.5** |

Table 4: Related language translation results measured in BLEU. The results for sr vs bs in measured in BLEU1 while cn vs tw is measure in BLEU.

representation and on the hidden states of the decoders to ensure that they have the same distribution. These are the two models that we mainly compare with. Prabhumoye et al. (2018) use the back-translation technique in their model, which is complementary to our method and can be integrated into our model to further improve performance. Gomez et al. (2018) use GAN-based approach to decipher shift ciphers. (Lample et al., 2017; Artetxe et al., 2017) propose unsupervised machine translation and use adversarial training to match the encoder representation of the sentences from different languages. They also use back-translation to refine their model in an iterative way.

**GANs**: GANs have been widely explored recently, especially in computer vision (Zhu et al., 2017; Chen et al., 2016b; Radford et al., 2015; Sutton et al., 2000; Salimans et al., 2016; Denton et al., 2015; Isola et al., 2017). The progress of GANs on text is relatively limited due to the non-differentiable discrete tokens. Lots of papers (Yu et al., 2017; Che et al., 2017; Li et al., 2017; Yang et al., 2017a) use REINFORCE (Sutton et al., 2000) to finetune a trained model to improve the quality of samples. There is also prior work that attempts to introduce more structured discriminators, for instance, the energy-based GAN (EBGAN) (Zhao et al., 2016) and RankGAN (Lin et al., 2017). Our language model can be seen as a special energy function, but it is more complicated than the auto-encoder used in (Zhao et al., 2016) since it has a recurrent structure. Hu et al. (2018a) also proposes to use structured discriminators in generative models and establishes its the connection with posterior regularization.

**Computer vision style transfer**: Our work is also related to unsupervised style transfer in computer vision (Gatys et al., 2016; Huang and Belongie, 2017). (Gatys et al., 2016) directly uses the covariance matrix of the CNN features and tries to align the covariance matrix to transfer the style. (Huang and Belongie, 2017) proposes adaptive instance normalization for an arbitrary style of images. (Zhu et al., 2017) uses a cycle-consistency loss to ensure the content of the images is preserved and can be translated back to original images.

**Language model for reranking**: Previously, language models are used to incorporate the knowledge of monolingual data mainly by reranking the sentences generated from a base model such as (Brants et al., 2007; Gulcehre et al., 2015; He et al., 2016). (Liu et al., 2017; Chen et al., 2016a) use a language model as training supervision for unsupervised OCR. Our model is more advanced in using language models as discriminators in distilling the knowledge of monolingual data to a base model in an end-to-end way.

## 6 Conclusion

We showed that by using language models as discriminators and we could outperform traditional binary classifier discriminators in three unsupervised text style transfer tasks including word substitution decipherment, sentiment modification and related language translation. In comparison with a binary classifier discriminator, a language model can provide a more stable and more informative training signal for training generators. Moreover, we empirically found that it is possible to eliminate adversarial training with negative samples if a structured model is used as the discriminator, thus pointing one possible direction to solve the training difficulty of GANs. In the future, we plan to explore and extend our model to semi-supervised learning.

## Footnotes

[1]We drop the subscript in notations wherever the meaning is clear.

[2]We use the code from `https://github.com/shentianxiao/language-style-transfer`.

[3]BLEU score is measured with `multi-bleu.perl`.

[4]`http://wortschatz.uni-leipzig.de/en`

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
