[Supplementary Material]

# A  Training Algorithms

---

**Algorithm 1** Unsupervised text style transfer.

---

**Input:** Data set of two different styles $\mathbf{X}$, $\mathbf{Y}$.
    Parameters: weight $\lambda$ and $\gamma$, temperature $\tau$.
    Initialized model parameters $\theta_{\mathbf{E}}, \theta_{\mathbf{G}}, \theta_{\text{LM}_{\mathbf{x}}}, \theta_{\text{LM}_{\mathbf{y}}}$.
    **repeat**
        Update $\theta_{\text{LM}_{\mathbf{x}}}$ and $\theta_{\text{LM}_{\mathbf{y}}}$ by minimizing $\mathcal{L}_{\text{LM}}^{\mathbf{x}}(\theta_{\text{LM}_{\mathbf{x}}})$ and $\mathcal{L}_{\text{LM}}^{\mathbf{y}}(\theta_{\text{LM}_{\mathbf{y}}})$ respectively.
        Update $\theta_{\mathbf{E}}, \theta_{\mathbf{G}}$ by minimizing: $\mathcal{L}_{\text{rec}} - \lambda(\mathcal{L}_{\text{LM}}^{\mathbf{x}} + \mathcal{L}_{\text{LM}}^{\mathbf{y}})$ using Equation 5.
    **until** convergence
**Output:** A text style transfer model with parameters $\theta_{\mathbf{E}}, \theta_{\mathbf{G}}$.

---

# B  Model Configurations

Similar model configuration to that of (Shen et al., 2017) is used for a fair comparison. We use one-layer GRU (Chung et al., 2014) as the encoder and decoder (generator). We set the word embedding size to be 100 and GRU hidden size to be 700. $\mathbf{v}$ is a vector of size 200. For the language model, we use the same architecture as the decoder. The parameters of the language model are not shared with parameters of other parts and are trained from scratch. We use a batch size of 128, which contains 64 samples from $\mathbf{X}$ and $\mathbf{Y}$ respectively. We use Adam (Kingma and Ba, 2014) optimization algorithm to train both the language model and the auto-encoder and the learning rate is set to be the same. Hyper-parameters are selected based on the validation set. We use grid search to pick the best parameters. The learning rate is selected from $[1e-3, 5e-4, 2e-4, 1e-4]$ and $\lambda$, the weight of language model loss, is selected from $[1.0, 0.5, 0.1]$. Models are trained for a total of 20 epochs. We use an annealing strategy to set the temperature of $\tau$ of the Gumbel-softmax approximation. The initial value of $\tau$ is set to 1.0 and it decays by half every epoch until reaching the minimum value of 0.001.

# C   Sentiment Transfer Examples

| Model | Negative to Positive |
|---|---|
| Original | it was super dry and had a weird taste to the entire slice . |
| (Shen et al., 2017) | it was super friendly and had a nice touch to the same . |
| (Hu et al., 2017a) | it was super well-made and had a weird taste to the entire slice . |
| LM | it was very good , had a good taste to the food service . |
| LM + classifier | it was super fresh and had a delicious taste to the entire slice . |
| | |
| Original | my goodness it was so gross . |
| (Shen et al., 2017) | my server it was so . |
| (Hu et al., 2017a) | my goodness it was so refreshing . |
| LM | my food it was so good . |
| LM + classifier | my goodness it was so great . |
| | |
| Original | maintenance is equally incompetent . |
| (Shen et al., 2017) | everything is terrific professional . |
| (Hu et al., 2017a) | maintenance is equally hilarious . |
| LM | maintenance is very great . |
| LM + classifier | maintenance is equally great . |
| | |
| Original | if i could give them a zero star review i would ! |
| (Shen et al., 2017) | if i will give them a breakfast star here ever ! |
| (Hu et al., 2017a) | if i lite give them a sweetheart star review i would ! |
| LM | if i could give them a _num_ star place i would . |
| LM + classifier | if i can give them a great star review i would ! |

| Model | Positive to Negative |
|---|---|
| Original | did n't know this type cuisine could be this great ! |
| (Shen et al., 2017) | did n't know this old food you make this same horrible ! |
| (Hu et al., 2017a) | did n't know this type cuisine could be this great ! |
| LM | did n't know this type , could be this bad . |
| LM + classifier | did n't know this type cuisine could be this horrible . |
| | |
| Original | besides that , the wine selection they have is pretty awesome as well . |
| (Shen et al., 2017) | after that , the quality prices that does n't pretty much well as . |
| (Hu et al., 2017a) | besides that , the wine selection they have is pretty borderline as atrocious . |
| LM | besides that , the food selection they have is pretty awful as well . |
| LM + classifier | besides that , the wine selection they have is pretty horrible as well . |
| | |
| Original | uncle george is very friendly to each guest . |
| (Shen et al., 2017) | if there is very rude to our cab . |
| (Hu et al., 2017a) | uncle george is very lackluster to each guest . |
| LM | uncle george is very rude to each guest . |
| LM + classifier | uncle george is very rude to each guest . |
| | |
| Original | the food is fresh and the environment is good . |
| (Shen et al., 2017) | the food is bland and the food is the nightmare . |
| (Hu et al., 2017a) | the food is atrocious and the environment is atrocious . |
| LM | the food is bad , the food is bad . |
| LM + classifier | the food is bland and the environment is bad . |

Table 5: Sentiment transfer examples.