[Reviews · NeurIPS 2018]

Reviewer 1



The paper proposes a new way for style-based transfer of text: using a language model as structured discriminator. Previous work using adversarial training to learn disentangled representations often results in non-fluent output and unstable training, two issues this paper addresses. The idea to use a language model and hence a non-binary discriminator is very nice, and is novel as far as I can tell. It it a good addition to recently proposed methods that do not require parallel text. The idea has the potential to be more generally applicable. The paper is well written and clear, particularly in the introduction. The experimental section though should be checked, as it contains several grammar mistakes (see below). I have two questions: 1. Decipherment results. Why are the results in Table 1--referred to as (Shen et al., 2017)--lower than those reported in the original paper? In particular, in the 60% and 80% setup, the reported results largely differ from what they report, resulting in the proposed model to fall *below* their results. Can you elaborate? 2. It is nice to see work on closely-related languages. However, why taking automatically translated data? There exists data for Croatian and Serbian, for instance (see references in http://www.aclweb.org/anthology/W16-4806). Using existing data would make the results more convincing. It would be nice if the paper could give some more details on how the classifier is integrated (line 243), as the LM alone is not sufficient to beat a simpler prior method (Li et al., 2018). minor nits (cf. lines): 74: two text dataset 182: in NLP literature 185: performs in task 186: token in vocabulary 200: improve the results 209: the whole sentences 215: to one-to-one mapping 242: the best of both model 273: missing space 277: are use 315: normalize for 321: a based model === Update after author response === Thanks for the insightful answers, particularly on my question #1.

Reviewer 2



The authors proposed a language model based discriminator in style transfer. First, the language model has the ability to judge the style of a sentence. Also, the continuous approximation makes the training stable by token-level supervision. Experiment results show that this model outperforms the baseline models. Strong points 1). The motivation of every component is well explained, and it is helpful in understanding the paper. 2). The continuous approximation is useful and interesting in supervising the generator by token-level. 3). The experiments are relatively sufficient, including the three tasks and the evaluation metrics. Weak points 1). The continuous approximation is one of your contributions. You should give more detailed analysis or visualization in this respect. For example, an ablation test without continuous approximation is needed (with Reinforce Algorithm). Typo: it is possible to “forgoe” adversarial As described in the conclusion, you plan to explore the semi-supervised learning, did you mean the semi-supervised style transfer? Summary This paper treats the GAN style transfer in a novel view: language model as the discriminator. And the proposed model provides token-level supervision signal to the generator, which makes the training stable. This paper not only promotes the style transfer task but also explores how to make the GAN training stable.

Reviewer 3



This paper proposes using the language models' log-likelihood score as a discriminator in order to train style transfer generation models. In particular, given a text x, the goal is to train an encoder and a decoder. The encoder extracts vectors z_x and v_x from x, representing the content and the style of x respectively. The decoder should be able to restore x from (z_x, v_x), but generate a stylistically different sentence from z_x and a different vector v_y. The paper proposes to judge the style difference by a language model through its log-likelihood score. Despite the simple idea, the authors show that it works well on three style transfer tasks, and achieves comparable or better performances than the state-of-the-art adversarially trained models. The results look promising. I agree with the authors that using language model as the discriminator is a compelling approach because it avoids the adversarial training's instability problem. It was mentioned in the paper that the language model discriminator has its own problems. One of them is that the language model always prefers short sentences. Thus additional length normalization and training hacks are necessary to make it work. The adversarially trained models don't seem to have this problem. A bigger concern of mine is that the proposed style transfer method could lose the semantics of the sentence more easily than the method of Shen et al. If x is a sentence from corpus X and y' is a sentence from corpus Y whose style is transferred to X. Shen et al.'s method requires the distributions of x and y' to be indistinguishable. In this paper, there is a penalty if y' has the same distribution as y, but there is no penalty if y' has a completely different distribution from that of x. In the extreme case, the total loss can be very low if the decoder learns to restore all the original input by memorizing and copying them, but always generates random strings if the content is paired with a transferred style vector. This doesn't happen in the paper's reported experiments, but I am worried that for harder tasks, given the strong overfitting capability of neural networks, the network would find it easier to learn such a copy-or-random generator than learning the actually useful style transfer model. Overall the paper is clearly written and has throughout experiments. Therefore I tend to accept the paper.